# Glutathione S-Transferase M3 Is Associated with Glycolysis in Intrinsic Temozolomide-Resistant Glioblastoma Multiforme Cells

**DOI:** 10.3390/ijms22137080

**Published:** 2021-06-30

**Authors:** Shu-Yu Cheng, Nan-Fu Chen, Zhi-Hong Wen, Zhi-Kang Yao, Kuan-Hao Tsui, Hsiao-Mei Kuo, Wu-Fu Chen

**Affiliations:** 1Department of Marine Biotechnology and Resources, National Sun Yat-sen University, No. 70, Lianhai Road, Gushan District, Kaohsiung City 804, Taiwan; joygetit@gmail.com (S.-Y.C.); wzh@mail.nsysu.edu.tw (Z.-H.W.); akang329@vghks.gov.tw (Z.-K.Y.); Hsiaomeikuo@gmail.com (H.-M.K.); 2Department of Environmental Protection, Green Technology Research Institute, CPC Corporation. No. 2, Zuonan Rd., Nan-Tzu District, Kaohsiung City 811, Taiwan; 3Department of Surgery, Division of Neurosurgery, Kaohsiung Armed Forces General Hospital, No. 2, Zhongzheng 1st Road, Lingya District, Kaohsiung City 802, Taiwan; k802chen@gmail.com; 4Institute of Medical Science and Technology, National Sun Yat-Sen University, No. 70, Lianhai Road, Gushan District, Kaohsiung City 804, Taiwan; 5Center for General Education, Cheng Shiu University, No. 840, Chengqing Road, Niaosong District, Kaohsiung City 833, Taiwan; 6Department of Orthopedics, Kaohsiung Veterans General Hospital, No. 386, Dazhong 1st Rd., Zuoying Dist., Kaohsiung City 813, Taiwan; 7Department of Obstetrics and Gynecology, Kaohsiung Veterans General Hospital, No. 386, Dazhong 1st Rd., Zuoying Dist., Kaohsiung City 813, Taiwan; khtsui60@gmail.com; 8Center for Neuroscience, National Sun Yat-Sen University, Kaohsiung 804, Taiwan. No. 70, Lianhai Road, Gushan District, Kaohsiung City, 804, Taiwan; 9Department of Neurosurgery, Kaohsiung Chang Gung Memorial Hospital and Chang Gung University College of Medicine, No. 123, Dapi Road, Niaosong District, Kaohsiung Cit 833, Taiwan; 10Department of Neurosurgery, Xiamen Chang Gung Hospital, 123 Xiafei Rd, Haicang District, Xiamen 361126, China

**Keywords:** glutathione S-transferase M3, glycolysis, chemotherapeutic resistance, TMZ

## Abstract

Glioblastoma multiforme (GBM) is a malignant primary brain tumor. The 5-year relative survival rate of patients with GBM remains <30% on average despite aggressive treatments, and secondary therapy fails in 90% of patients. In chemotherapeutic failure, detoxification proteins are crucial to the activity of chemotherapy drugs. Usually, glutathione S-transferase (GST) superfamily members act as detoxification enzymes by activating xenobiotic metabolites through conjugation with glutathione in healthy cells. However, some overexpressed GSTs not only increase GST activity but also trigger chemotherapy resistance and tumorigenesis-related signaling transductions. Whether GSTM3 is involved in GBM chemoresistance remains unclear. In the current study, we found that T98G, a GBM cell line with pre-existing temozolomide (TMZ) resistance, has high glycolysis and GSTM3 expression. GSTM3 knockdown in T98G decreased glycolysis ability through lactate dehydrogenase A activity reduction. Moreover, it increased TMZ toxicity and decreased invasion ability. Furthermore, we provide next-generation sequencing–based identification of significantly changed messenger RNAs of T98G cells with GSTM3 knockdown for further research. GSTM3 was downregulated in intrinsic TMZ-resistant T98G with a change in the expression levels of some essential glycolysis-related genes. Thus, GSTM3 was associated with glycolysis in chemotherapeutic resistance in T98G cells. Our findings provide new insight into the GSTM3 mechanism in recurring GBM.

## 1. Introduction

Glioblastoma multiforme (GBM) is the most aggressive primary brain cancer and is typically associated with a poor prognosis. The 5-year relative survival rate of patients with GBM who receive surgical resection combined with radiation therapy and chemotherapy is approximately 5% [1]. Temozolomide (TMZ) is currently the standard chemotherapy drug for patients with primary GBM approved by the US Food and Drug Administration [2,3]. However, strong intrinsic resistance to chemotherapy is one of the main reasons for GBM recurrence [4]. Therefore, investigating the molecular components involved in chemotherapy response alteration leading to cancer therapy failure is crucial to understand cancer recurrence.

Multidrug resistance (MDR) is a series of cellular drug metabolism processes that are majorly responsible for chemotherapeutic drug resistance [5]. Glutathione s-transferase (GST) is a detoxification enzyme superfamily of electrophilic xenobiotics involved in MDR [6]. The GST superfamily consists of mitochondrial [one class: κ (K)], microsomal [one class: MGST], and cytosolic [seven classes: α (A), µ (M), ω (O), σ (S), θ (T), π (P) and ζ (Z)] GSTs. GST classes are identified on the basis of sequence similarity, immune cross-reactivity, and substrate specificity [7,8,9].

Notably, elevated levels of GSTs and glutathione are highly associated with cancer chemotherapeutic drug resistance [10]. Some GST superfamily members play a detoxification role in cancer. GSTP1 interacts with Jun N-terminal kinase (JNK) and inhibits apoptosis in lung cancer cells [11] and gastric cancer cells [12,13]. GSTO1 inhibits JNK signaling and reduces autophagy to block cisplatin-induced cervical cancer cells [14] and aflatoxin B1-induced cell programmed death in macrophages [15]. GSTM2 prevented programmed cell death of cancer cells through autophagy inhibition and lysosome dysfunction in an aminochrome-treated GBM cell line [16]. These findings indicate that the GST superfamily has many unknown functions in addition to MDR-related processes in chemotherapy. However, to our knowledge, the association of GSTM subfamily members with the glycolytic pathway in GBM has not yet been investigated.

Glycolysis is one of the vital processes in cellular respiration and converts glucose into energy in universal cells. In tumors, abnormally consistent release of adenosine triphosphate (ATP) and lactate through glycolysis under aerobic conditions is called the Warburg effect [17,18]. The Warburg effect involves rapid ATP synthesis, biosynthesis enhancement, microenvironment acidification, and reactive oxygen species generation to mediate cellular signaling for aggressive tumor growth [19,20]. Furthermore, it is an essential feature in GBM tumors [21,22,23,24] and is highly correlated with chemotherapeutic drug resistance [25]. In aerobic glycolysis, lactate dehydrogenase A (LDHA) is crucial for catalytic conversion of pyruvate to lactate and back. Moreover, LDHA is involved in angiogenesis, immune escape, metastasis, and proliferation in the tumor generation process [26]. Blocking the LDHA glycolytic pathway increases the sensitivity of GBM cells to radiation and TMZ [27]. Hence, altered glycolytic metabolism is highly correlated with improved prognosis and can act as a target for cancer therapy [28]. In this paper, we discuss the correlation between GSTM subfamily proteins and glycolysis in GBM cells.

In the current study, we analyzed GSTM subfamily members’ protein expression in GBM cell lines. We found that GSTM3 is highly expressed and affects glycolysis in TMZ-resistant T98G cells. Moreover, next-generation sequencing (NGS) analysis demonstrated the changing RNA profile of GSTM3 RNA-inhibited T98G cells. In this study, the T98G GBM cell line with pre-existing TMZ resistance was used to investigate the novel cellular function and related gene transcriptions of GSTM3.

## 2. Results

### 2.1. TMZ Resistance Level and Prognostic Marker Expression in GBM Cell Lines

Currently, TMZ is the most used chemotherapeutic drug for GBM treatment. However, most patients with GBM still have poor prognosis after the first treatment. The TMZ-resistance levels of commonly used and purchasable GBM cell lines were compared. T98G and U138MG are TMZ-resistant GBM cell lines, and U87 and A172 are TMZ-sensitive GBM cell lines [29]. However, TMZ treatment involves T98G, U138MG, A172, GBM8401, and U87MG cells for 72 h. In the 3-(4,5-dimethylthiazol-2-yl)-2,5-diphenyltetrazolium bromide (MTT) assay, the toxicity response of those cell lines was as follows: T98G < U138MG < A172 < GBM8401 < U87MG (Figure 1). Therefore, T98G was the most resistant to TMZ among the studied GBM cell lines. Statistical analysis demonstrated that the T98G cell line was the most resistant to TMZ. We performed one-way analysis of variance and a post hoc test between different cell lines using the same TMZ concentration.

### 2.2. GST Activity and GST Subfamily Protein Expression in GBM Cell Lines

Detoxification is one of the multiple factors involved in chemotherapeutic drug and radiotherapy resistance. The GST superfamily is one of the largest enzyme groups involved in the detoxification process of living cells. The GST activity assay of five GBM cell lines revealed detoxification in the following order: T98G > A172 > U87MG > U138MG > GBM8401 (Figure 2A). In malignant glioma, the expression of GSTP1 protein, the most discussed GST superfamily member, is increased and contributes to TMZ resistance [30]. However, few studies have discussed the GSTM subfamily in GBM. In this study, we evaluated the protein expression of some GST superfamily members in the human fetal glial cell line SVGp12 and GBM cell lines T98G, U138MG, A172, GBM8401, and U87MG (Figure 2B). Western blot analysis revealed that cytosolic GSTP1 and mitochondrial GSTK1 proteins were highly expressed in the aforementioned cell lines. However, the expression of GSTM subfamily proteins was lower in the SVGp12 cell line than in the GBM cell lines. Furthermore, GSTM3 protein expression in T98G was higher than in the other four GBM cell lines. Therefore, GSTM3 may play a role in the TMZ resistance of T98G.

### 2.3. Glycolysis Stress Response of Well Known GBM Cell Lines

To obtain an overall picture of the bioenergetics profile of the five well known GBM cell lines studied, we plotted the basal extracellular acidification rate (ECAR) against the mitochondrial oxygen consumption rate (OCR), with data obtained using a Seahorse Extracellular Flux (XF) Analyzer (Figure 3a). The bioenergetics profile showed that T98G and A172 cells were the most energetic when compared with the other three GBM cell lines. Generally, tumors prefer the glycolysis metabolic pathway for rapid energy, which is described as the Warburg effect [31]. This cellular response is one of the tumorigenesis factors in cancer development that causes not only hypoxia but also normoxic conditions [32]. Additionally, the Warburg effect affects drug efficacy in cancer therapy [33]. In glycolysis, lactic acid dissociates into lactate anions and protons and is released into the extracellular medium, a state that is directly measured as the ECAR as an indirect readout of the cellular glycolytic flux [34]. Therefore, we used the glycolysis stress test kit with the acute injection protocol and Seahorse XF Analyzer for validating the real-time glycolysis stress level of GBM cells. First, glucose was added to reach the saturation level after cell starvation, and during this period, the ECAR represented the glycolytic rate. Subsequently, oligomycin, an inhibitor of mitochondrial oxidative phosphorylation, was added for glycolysis. During this process, the ECAR represented the glycolytic capacity of the cell. Next, 2-DG was added to halt glycolysis to measure the maximum glycolysis ability of the cell, indicating the glycolytic reserve. In this study, we evaluated the glycolysis stress levels of the T98G, U138MG, A172, GBM8401, and U87MG cell lines (Figure 3b).

The statistics of the glycolysis stress levels of these five GBM cells showed that T98G had the highest glycolytic rate, T98G and A172 had the highest glycolytic capacity, and A172 had the highest glycolytic reserve (Figure 3c). These results suggest that TMZ-resistant T98G and TMZ-sensitive A172 GBM cells have high glycolysis stress ability. However, endogenous high expression of GSTM3 was only observed in T98G among these five well-known GBM cell lines. Therefore, we evaluated the chemotherapeutic-resistance-related mechanism of GSTM3 by using T98G cells.

### 2.4. GSTM3 Affect TMZ Toxicity in T98G Cells

To understand the role of GSTM3 in the resistance to the chemotherapeutic drug TMZ, we used the small interfering RNA (siRNA) gene silencing method to downregulate GSTM3 in endogenous GSTM3-expressed T98G cells. Cells were knocked down for treating siRNA for 72 h, and the target gene GSTM3 had a considerable silencing effect (Figure 4a). Microscope observation revealed no visible change in the morphology of GSTM3-knockdown (siGSTM3) T98G cells compared with nontarget control (siNC) T98G cells (Figure 4b). Additionally, the GST activity was significantly decreased (by 20%) after GSTM3 knockdown (Figure 4c). Subsequently, the cell toxicity of TMZ in siGSTM3 was considerably higher compared with that in siNC T98G GBM cells (Figure 4d). These results indicate that GSTM3 affects TMZ resistance in T98G GBM cells with pre-existing TMZ resistance.

### 2.5. GSTM3 Associated with Glycolysis in T98G Cells

The Warburg effect, hypoxia, and abnormal angiogenesis are associated with the progression of GBM tumors [35]. To investigate the metabolic effect of GSTM3, we assessed real-time mitochondrial and glycolysis stress in T98G by using the Seahorse XF Analyzer. The cellular OCR was evaluated as mitochondrial stress. The OCR value did not considerably change after the sequential addition of mitochondria-perturbing agents oligomycin, FCCP, and rotenone/antimycin A in GSTM3-knockdown T98G cells (Figure 5a). However, the sequential addition of glycolysis-perturbing agents glucose, oligomycin, and 2-DG revealed that the ECAR of GSTM3-knockdown cells was significantly lower than that of the negative control of T98G cells (Figure 5b). GSTM3 knockdown significantly decreased the glycolytic rate, glycolytic capacity, and glycolytic reserve of T98G cells (Figure 5c). In tumors, lactic acid dehydrogenase (LDHA) is a crucial protein for the conversion of pyruvate to lactic acid, which causes an aggressive glycolytic cancer phenotype, resulting in the Warburg effect [36,37]. LDH inhibition suppressed tumor progression in a preclinical tumor model [38]. In the current reverse-transcription polymerase chain reaction (RT-PCR) result, LDHA messenger RNA (mRNA) expression did not change after *gstm3* expression inhibition (Figure 6a). However, the activity of LDH (Figure 6b) and glycolysis final product L-lactate was significantly decreased in GSTM3 gene inhibition T98G. Nevertheless, in GBM cells, LDHA as an oncogene affects cancer cell movement in A172 and U87MG cells [20]. Moreover, high invasion and metastasis ability is one of the recognizable characteristics and a treatment target for therapy in malignant tumors of the central nervous system [39]. Our data showed that GSTM3 affected LDHA activity in T98G cells. We expected GSTM3 downregulation to also decrease cell invasion. In the Transwell invasion assay, we observed that the invasion closure rate of GSTM3 knockdown in T98G cells was significantly lower than that in siNC T98G cells (Figure 7a). The invasion ability was decreased by 40% in GSTM3 knockdown cells (Figure 7b). These results indicated that GSTM3 downregulation interfered with glycolysis through LDHA activity inhibition and invasion ability in T98G.

### 2.6. Transcriptomic Analyses of Gene Expression Through siRNA Knockdown of GSTM3 in TMZ-Resistant T98G Cells

To investigate the genes affected by GSTM3, we used NGS technology to evaluate the alteration of the whole mRNA profile in GSTM3-knockdown T98G cells. In total, 33 upregulated and 35 downregulated genes significantly changed (*p* < 0.05) twice during mRNA transcription in GSTM3 knockdown cells compared with siNC T98G cells (Figure 8). The five most downregulated genes in GSTM3-knockdown T98G cells were GRB10 interacting GYF protein 2 (GIGYF2), tryptophan-transfer RNA ligase (WARS), dynein cytoplasmic 1 intermediate chain 2 (DYNC1I2), 26S proteasome complex subunit (SEM1 or Dss1), and major facilitator superfamily domain containing 1 (MFSD1; Table 1). Furthermore, the five most upregulated (Table 2) genes in GSTM3-knockdown T98G cells were follistatin-like 3 (FSTL3); dynactin subunit 2 (DCTN2); proteolipid protein 1 (PLP1); SWI/SNF-related, matrix-associated, actin-dependent regulator of chromatin, subfamily a, member 2 (SMARCA2); and septin 9 (SEPT9). Among them, global transcription activator, encoded by SMARCA2, was reported to have a potential tumor suppressor activity in cancer [40]. SEPT9 gene downregulation could be a tumorigenesis marker in colon tissues [41]. Furthermore, WRS could be a therapeutic target in cancer [42]. SEM1 overexpression is highly correlated with poor prognosis in patients with melanoma and squamous cell carcinoma [43]. To the best of our knowledge, information regarding FSTL3, DCTN2, PLP1, GIGYF2, DYNC1I2, and MFSD1 related to cancer has been negligible until now. In summary, the mRNA profile results of NGS analysis in T98G cells demonstrated that GSTM3 may be associated with tumorigenesis and improved prognosis.

## 3. Discussion

In systemic resistance, metabolism is essential for directly destroying the toxic substrate inside the cell. In the metabolism process, the phase I metabolic enzyme cytochrome P450 (CYP) enzymes oxidize the compound into xenobiotic groups, and then, phase II metabolic enzyme GSTs conjugate CYP-oxidized cytoplasmic metabolites into inactive molecules [44]. Therefore, the GST superfamily is one of the crucial functional protein groups involved in the detoxification and systemic resistance of drugs in cells.

Studies have shown GST overexpression in many cancer types. The most investigated GST family protein, GSTP1, was enhanced in ovarian [45], colonic [46,47], lung [48], and gastric [49] tumors relative to normal tissues. In human squamous cell carcinoma, pathological observations indicated a strong connection between higher GSTP1 nuclear staining and shorter survival [50]. Moreover, GSTM3 and GSTP1 are involved in cell survival and proliferation in cervical cancer cell lines [51], but information is not available regarding their function in GBM. Furthermore, GST overexpression has been demonstrated to be correlated with cancer drug resistance. Increased GSTP was observed in inhibitor of growth-5-mediated chemotherapeutic resistance in the SGC-7901 human gastric cancer cell line [52]. MCF-7 cells with doxorubicin-induced resistance were found to overexpress GSTP RNA [53]. An in vitro study demonstrated that enzyme activity in a GSTP1 variant, Val105, was reduced by 80% [54], and this variant exhibited better survival than the wild type genotype in patients with non-small-cell lung carcinoma after platinum-based chemotherapy [55,56]. Moreover, a proteomics analysis indicated that GSTM3, GSTP1, GSTO1, GSTK1, and microsomal GST1 are significantly increased in Adriamycin-resistant MCF-7 cells [57]. Furthermore, some direct evidence indicates that GSTs are involved in drug and radiation resistance in cancers. For example, GSTP1 affects the toxicity efficiency of docetaxel and paclitaxel in breast cancer cells [58], platinum drugs in ovarian cancer cells [59], and ionizing radiation in prostate carcinoma cells [60]. GSTA1 mediates cisplatin in some solid cancer cells [61] and imatinib in leukemia cells [62]. Recently, treatment with micro RNA-133b-silenced GSTP1 with chemotherapy agent cisplatin or paclitaxel treatment reduced cell viability in epithelial ovarian cancer [63]. However, little information is available regarding GSTM subfamily members in GBM.

The large GSTM subfamily in GST includes five members. Although the gene locations of the five GSTM subfamily members are close to each other on chromosome 1, the transcription of gene GSTM3 proceeds in a different direction to that of the other members. In this study, we discovered the GSTM subfamily members’ expression to be different in five well known GBM cell lines (Figure 3b). Interestingly, GSTM subfamily member GSTM2 mimics GSTM1 in case of GSTM1 homozygous deletion to compensate for GST catalytic activity [64]. Moreover, phylogenetic analyses revealed that the GSTM3 gene diverges from other GSTM genes in early evolution compared with human-related primate species [65]. In T98G cells, GSTM3 protein expression is considerably higher than that of other GSTM subfamily members (Figure 3b). Therefore, we speculate that the random appearance of expression of GSTM subfamily members might be due to adaptations to complement insufficient GST activity, which is affected by altered GSTM members. Moreover, we found that GSTM3 not only affects TMZ toxicity (Figure 4d) but also inhibits LDHA activity, leading to decreased glycolysis (Figure 5b,c) in T98G cells. The results indicated that each GSTM subfamily member not only plays a role in detoxification but is also involved in other cellular functions in cancer. Recently, a report indicated long noncoding RNA *Homo sapiens* GST mu 3, transcript variant 2, noncoding RNA (GSTM3TV2) promotes pancreatic cancer gemcitabine resistance by increasing L-type amino acid transporter 2 and oxidized low-density lipoprotein receptor 1 though competitive disruption of micro RNA let-7. Moreover, high levels of GSTM3TV2 expression in tumor samples led to significantly shorter overall survival than low levels of GSTM3TV2 expression in tumors in patients with pancreatic cancer [66]. Therefore, improved understanding of the functions of the GSTM subfamily in different cancers can provide possible solutions to drug resistance in cancer.

Some reports have indicated the role of GSTM subfamily members in the normal function and diseases of the central nervous system. The tissue distribution of GST subunits in humans shows that cytosolic GSTM3 is highly expressed and most active in total GST activity in the human testis and brain [67,68]. The high-expression GSTM3 metabolism network is strongly corelated to amino acid catalysis but weakly to glutathione metabolism during cell differentiation induction in the human left cerebrum [68]. GSTM3 is probably related to sleep–wake regulation, nociception, and temperature regulation because this enzyme catalyzes the conversion of cytosolic prostaglandin H2 into prostaglandin E2 in the brain [69]. Moreover, regarding Alzheimer’s disease (AD), a multifactorial aging-related brain disorder, decreased transcript levels of *gstm3* in AD have been reported to affect hippocampus and peripheral blood mononuclear cells [70,71]. Notably, a report indicated that the GSTM3 single nucleotide polymorphism rs7487 alters enzyme activity in patients with early onset AD [72,73]. Additionally, a report indicated that GSTM2 inhibits the aminochrome formation through the conjugation of dopamine o-quinone to 5-glutathionyl-dopamine in the brain [74]. According to these reports, GSTM subfamily members may be involved in some of the complex brain functions, and further research is needed to understand the comprehensive role of each member in brain disease.

The Warburg effect, altered glycolytic metabolism, is highly correlated with cancer prognosis and is a target for cancer therapy [28]. Many glycolysis enzymes are known to be related to chemotherapy drug resistance in several cancers. Accordingly, identifying molecules associated with glycolysis can help improve treatment strategies for the Warburg effect in cancer. Some reports have indicated that glycolytic metabolism is associated with brain tumor treatment. For example, blocking the LDHA glycolytic pathway is one of the crucial methods of sensitizing GBM cells to radiation and TMZ [27]. Furthermore, glucose transporter inhibitors increase TMZ toxicity in high-grade gliomas [75]. Dichloroacetate, a pyruvate dehydrogenase kinase 1 inhibitor, reverses increased glycolysis activity and reduces TMZ resistance in glioblastoma [76]. Therefore, glycolytic enzyme regulation is an ongoing therapeutic strategy in drug design for cancer treatment. However, GSTP1 is the only GST superfamily member with reported effects on glycolysis. GSTP1 directly binds to glyceraldehyde-3-phosphate dehydrogenase (GAPDH) and activates its glycolytic enzyme function in triple-negative breast cancer cells, as demonstrated by pharmacological inactivation [77]. In this study, we found that in GSTM3 knockdown T98G dells, glycolysis was reduced and LDHA was significantly inactivated with (Figure 5 and Figure 6). GSTM3 may be one of the crucial detoxification protein families in chemotherapy-resistant GBM, and it is worthy of further research.

GBM is the most malignant brain cancer and is characterized by highly diffuse infiltration of tumor cells into surrounding healthy tissue, damaging neurological functions. Uncontrollable replication and invasion of GBM into the functional areas of the brain increase the difficulty of neurosurgical extirpation, and the corresponding extremely poor prognosis leads to early mortality [78]. Glycolysis is the primary bioenergetic pathway for cell motility and cytoskeletal remodeling in cancer cells [79]. Therefore, understanding the critical molecular mechanisms underlying brain cancer invasion is one of the primary therapeutic strategies for brain cancer. Studies using in vitro models have demonstrated that GST superfamily members affect cell invasion; for example, downregulated GSTP1 influences chemosensitivity and inhibits invasion ability in cisplatin-resistant ovarian cancer cells [59]. Decreased GSTA1 in A549 lung carcinoma affects epithelial–mesenchymal transition markers’ mRNA expression and decreases invasion ability [80]. In this study, we observed that GSTM3 knockdown reduced the invasion ability of T98G cells (Figure 7).

Additionally, we determined the upregulated and downregulated genes of inhibition of GSTM3 in intrinsically TMZ-resistant T98G cells by using NGS (Table 1 and Table 2). To the best of our knowledge, no RNA profile report has correlated to GSTM3-related transcripts until now. In GSTM3-inhibited T98G cells, upregulated genes SMARCA2 and SEPT9 were found to suppress tumor markers [40,41]. Furthermore, WRS and SEM1, downregulated genes in GSTM3 downregulation, showed poor correlation with treatment outcomes in patients with cancer [42,43]. We hope to provide more information on GST subfamily members. Overall, these studies suggest that GSTM3 not only causes detoxification but also induces glycolysis in cancer.

## 4. Methods

### 4.1. Reagents

MTT was purchased from SERVA (Heidelberg, Germany). TMZ, dimethyl sulfoxide (DMSO) and glucose were obtained from Sigma-Aldrich^®®^ (Merck KGaA, Darmstadt, Germany). Acridine orange, TRIzol Reagent, minimum essential medium eagle alpha modifications (alpha-MEM), Roswell Park Memorial Institute medium (RPMI) 1640, heat-inactivated fetal bovine serum (FBS), penicillin–streptomycin (P/S), pyruvate, and L-glutamate were purchased from Thermo Fisher Scientific (Waltham, MA, USA).

### 4.2. Antibodies

GSTM1 (1H4F2), GSTM2 (E-9), GSTM4 (PL-B12), and GSTK1 (E-4) were obtained from Santa Cruz Biotechnology (Dallas, TX, USA). β-actin Ab was purchased from Cell Signaling Technology (Danvers, MA, USA). GSTM3 Ab was procured from ProteinTech Group (Rosemont, IL, USA). GSTP1 and GSTM5 Ab were from Genetex (Irvine, CA, USA).

### 4.3. Cell Culture

The human brain malignant glioma cell lines U87MG and GBM8401 [81] were purchased from the Food Industry Research and Development Institute (Hsinchu, Taiwan). The human fetal astroglia cell line SVGp12 and human brain malignant glioma cell lines T98G, A172, and U138MG were purchased from the American Type Culture Collection (Manassas, VA, USA). Human GBM cell lines U87MG, U138MG, and T98G and SVGp12 astroglia were maintained in alpha-MEM; human GBM8401 glioblastoma cells were maintained in RPMI 1640; and human A172 glioblastoma cells were maintained in Dulbecco’s modified Eagle’s medium. All media contained 10% FBS and 50 U/mL P/S. Cells were cultured under a humid atmosphere consisting of 5% CO_2_ in 95% air at 37 °C. The cell lines were subcultured every 2–3 days up to the 10th passage. The cells were used in the following experiments.

### 4.4. MTT Assay for Cell Viability Analysis

The cell viability of the GBM cell lines was determined using an MTT assay. Cells at a density of 2 × 10^4^/well were seeded onto a 96-well plate. The next day, various concentrations of drugs were added. At the end of the exposure time, 20 µL of 5 mg/mL MTT was added to each well, which was then incubated at 37 °C for 4 h. The medium was carefully removed, and 200 µL of DMSO was added to each well. Absorbance at 570 nm was read by Epoch™, enzyme-linked immunosorbent assay reader (BioTek, Winooski, VT, USA).

### 4.5. Western Blotting

U87MG and GBM8401 cells were treated at the indicated time points of prodigiosin in various concentrations depending on the experiment. Supernatants were collected, and cells were washed with phosphate buffered saline (PBS) before the addition of RIPA lysis buffer containing complete ULTRA protease inhibitor cocktail tablets (Roche Diagnostics, Mannheim, Germany). Immunoreactive bands were visualized using Immobilon Western Chemiluminescent HRP Substrate (Merck Millipore, MA, USA). Images were obtained using the UVP BioChemi Imaging System (Analytik Jena GmbH, Jena, Germany) and relative densitometric quantification was performed using LabWorks 4.0 software (Analytik Jena GmbH, Jena, Germany).

### 4.6. Gene Knockdown of T98G Cells

siRNA was used to downregulate the target gene in this research. The siRNA transfections were performed using Lipofectamine RNAiMAX kit (Thermo Fisher #13778030) following the manufacturer’s instructions. The following siRNA sequences were provided by the Dharmacon part of GE Healthcare (Lafayette, CO, USA): nontargeting pool (UGGUUUACAUGUCGACUAA, UGGUUUACAUGUUUUCUGA, UGGUUUACAUGUUGUGUGA and UGGUUUACAUGUUUUCCUA); GSTM3 (UUACAGCUCUGACCACGAA, GGGAAAUUCUCAUGGUUUG, GAGUGGACAUCAUAGAGAA and CAACAUGUGUGGUGAGACU); and GAPDH (GUCAACGGAUUUGGUCGUA, CAACGGAUUUGGUCGUAUU, GACCUCAACUACAUGGUUU and UGGUUUACAUGUUCCAAUA).

### 4.7. GST Activity Assay

The GST activity of GBM cells was measured using a GST Assay Kit (catalog number: 703302, Cayman Chemical, Ann Arbor, MI, USA). Each sample (100 µg) was reacted at 25 °C with 0.1 M KPO_4_ buffer, pH 7.5, containing 1 mM GSH and 1 mM CDNB. Measurements were obtained at 340 nm by using a CLARIOstar Multimode Microplate Reader (BMG LABTECH, Ortenberg, Germany) every 5 min. GST activity was calculated using the following formula:GST activity = (Δ340/min)/0.00503 µM^−1^ × 0.2 mL/0.02 mL × Sample dilution(1)

### 4.8. LDH Activity Assay

The LDH activity of each siRNA-treated GBM cell line was measured using a Pierce LDH Cytotoxicity Assay Kit (Catalog Number: 88953, Thermo Scientific, Waltham, MA, USA). Each measurement was obtained for 1 × 10^5^ cells. The endogenous proteins of each sample were lysed with 50 µL of 1× lysis buffer and reacted with a reaction mixture for 30 min while protected from light. After incubation, 50 µL of stop solution was added, absorbance was measured at 490 nm, and the background signal was measured at 680 nm by using a CLARIOstar Multimode Microplate Reader. The final LDH activity value was obtained through subtraction of the 680 nm absorbance value from the 490 nm absorbance value.

### 4.9. L-Lactate Assay

The L-lactate concentration of each siRNA-treated GBM cell type was measured using an L-Lactate Assay Kit (Catalog Number: KA3776, Abnova, Taipei, Taiwan). First, 1 × 10^5^ cells were lysed using 100 µL of RIPA lysis buffer. Subsequently, 50 µL of each sample was added to separate wells of a black 96-well plate, which was followed by addition of a working reagent after mixing 40 µL of assay buffer, 1 µL of enzyme A, 1 µL of enzyme B, 10 µL of NAD, and 5 µL of the probe. The plates were incubated for 60 min at room temperature while protected from light. The fluorescent signal was read at λ_ex/em_ = 530/585 nm by using a CLARIOstar Multimode Microplate Reader. Lactate concentrations were calculated from the standard curve (ranging from 0 to 40 µM) generated from the lactate standard and in accordance with the manufacturer’s instructions.

### 4.10. Relative Quantification of Target Gene Expression through RT-PCR

Total cellular RNA was isolated using the TRIzol Reagent method (Invitrogen Life Technologies, Carlsbad, CA, USA). The amplification of cDNA was performed using iQTM SYBR Green Supermix (Bio-Rad Laboratories, Hercules, CA, USA) in 96-well low-profile PCR plates (Scientific Specialties Inc., Lodi, CA, USA) on the CFX96 TM Real-Time System (Bio-Rad). The reaction mixture (20 µL) contained 10 µL of 2× iQTM SYBR Green PCR Supermix (Bio-Rad™). The mRNA was amplified using gene-specific primers designed on the basis of available SYBR Green I fluorescence monitored using a real-time PCR system CFX96 TM (Bio-Rad™) equipped with the CFX Manager TM Software (Bio-Rad). The expression levels of target mRNAs were quantified in accordance with the expression level of the housekeeping gene GAPDH by using the 2(-Delta Delta C(T)) method. The primers used for RT-PCR were as follows: LDHA: 5′-TTGGTCCAGCGTAACGTGAAC-3′ and 5′-CCAGGATGTGTAGCCTTTGAG-3′; and GAPDH: 5′-GACCCCTTCATTGACCTCAAC-3′ and 5′-CTTCTCCATGGTGGTGAAGA-3′. The RT-PCR conditions were as follows: 95 °C for 5 min; 40 cycles of 95 °C for 30 s, 56 °C for 30 s and 72 °C for 50 s.

### 4.11. Live-Cell Metabolic Assay

We used a Seahorse XF 96 Analyzer (Seahorse Bioscience, Inc., North Billerica, MA, USA) to measure the OCR, an indicator of mitochondrial respiration, and the ECAR, an indicator of glycolysis, in real time in GBM cell lines. Twenty-four hours before the assay, cells were seeded onto an XF 24 cell culture microplate (Seahorse Bioscience) at the following GBM cell density in the culture medium: T98G and U138: 1 × 10^5^, U87MG: 1.5 × 10^5^ and GBM8401: 1.2 × 10^5^ cells/well. For ECAR analysis, the medium was replaced with Seahorse XF base medium (Seahorse Bioscience) supplemented with 1 mM L-glutamate at 37 °C 1 h before the ECAR analysis. In OCR analysis, the medium was replaced with Seahorse XF base medium supplemented with 1 mM pyruvate, 2 mM L-glutamate, and 10 mM glucose at 37 °C before 1 h. Cell metabolic stress responses were recorded through the addition of drugs and real-time recording by using the Seahorse XF 96 Analyzer. Cells were plated with at least three replicate wells for each test. Equations of glycolytic rate, glycolytic capacity and glycolytic reserve were referenced from Agilent Seahorse XF Glycolysis Stress Test Report Generator User Guide. Those were calculated using the following formula:Glycolytic rate = (maximum rate measurement before Oligomycin injection) − (last rate measurement before glucose injection)Glycolytic capacity = (maximum rate measurement after Oligomycin injection) − (last rate measurement before glucose injection)Glycolytic reserve = (glycolytic capacity) − (glycolytic rate)

### 4.12. Cancer Cell Invasion Assay

For the invasion assay, T98G cells (2 × 10^4^ cells/chamber) were seeded in 1% FBS in gelatin-coated upper 8-µm pore Transwell chambers (Corning Inc., Corning, NY, USA) and in 10% FBS in the lower chambers to induce cell migration. After 24 h, the cells were washed with 1× PBS, fixed with prechilled methanol for 2 h, and stained for 24 h with crystal violet. The contents of the upper Transwell chambers were wiped off using a cotton swab, and cells in the lower chambers were imaged through Leica DMI 3000B phase-contrast microscopy (Leica Camera, Wetzlar, Germany).

### 4.13. RNA Profile Transcriptome Based on NGS

RNA expression profiles of siRNA-treated T98G cells were analyzed using NGS. The NGS transcriptome sequencing and data analysis were performed by Welgene Biotech Co., Ltd. (Taipei, Taiwan). After 72 h of GSTM3 siRNA treatment, RNA was extracted using TRIzol Reagent (Invitrogen, USA) in accordance with the instruction manual. Purified RNA was quantified at OD_260_ by using an ND-1000 spectrophotometer (NanoDrop Technologies LLC, Wilmington, DE, USA) and quantitated using a Bioanalyzer 2100 (Agilent Technology, Santa Clara, CA, USA) with the RNA 6000 LabChip kit (Agilent Technology, Santa Clara, CA, USA). All RNA sample preparation procedures were performed in accordance with Illumina’s official protocol. Agilent’s SureSelect Strand-Specific RNA Library Preparation Kit was used for library construction, and AMPure XP beads (Beckman Coulter, Brea, CA, USA) were then used for size selection. The sequences were determined using Illumina’s sequencing-by-synthesis technology (Illumina, San Diego, CA, USA). Sequencing data (FASTQ reads) were generated using Welgene Biotech’s pipeline by using Illumina’s base-calling program bcl2fastq version 2.20. Basecalls were converted using the Illumina official tool, bcl2fastq2 conversion Software version 2.19, which is used to convert BCL files from all Illumina sequencing systems. Both adaptor clipping and sequence quality trimming of Illumina FASTQ data were accomplished using Trimmomatic version 0.32.

## 5. Conclusions

In this research, we obtained experimental results from an initial investigation of the GSTM subfamily. We discovered that GSTM subfamily members are randomly ex-pressed in GBM cell lines. However, we found intrinsic TMZ-resistant T98G cells to have the highest glycolysis rate (Figure 2) and GST activity (Figure 3a). Furthermore, GSTM3 protein expression is high in T98G cells (Figure 3b). Knockdown of GSTM3 enhances TMZ toxicity (Figure 4b) and decreases glycolysis (Figure 5 and Figure 6) and cell invasion in T98G GBM cells with pre-existing TMZ resistance (Figure 8). These findings demonstrate that GSTM3 not only provides detoxification ability but is also involved in glycolysis-related cellular behavior.

## Figures and Tables

**Figure 1 ijms-22-07080-f001:**
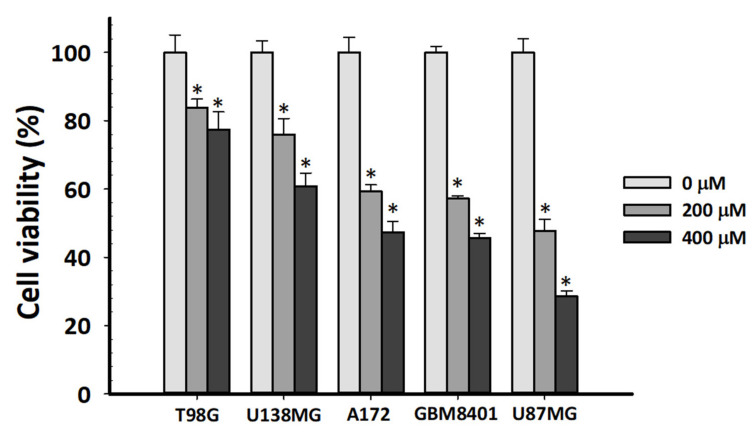
Response of glioblastoma cell lines to chemotherapy drug temozolomide (TMZ). We performed 3-(4,5-dimethylthiazol-2-yl)-2,5-diphenyltetrazolium bromide (MTT) assays on T98G, U138MG, A172, GBM8401 and U87MG cell lines after 72 h of nontreatment or 200 or 400 µM of TMZ treatment. Values are expressed as mean ± standard deviation (SD; *n* = 4). * *p* < 0.05 relative to nontreatment.

**Figure 2 ijms-22-07080-f002:**
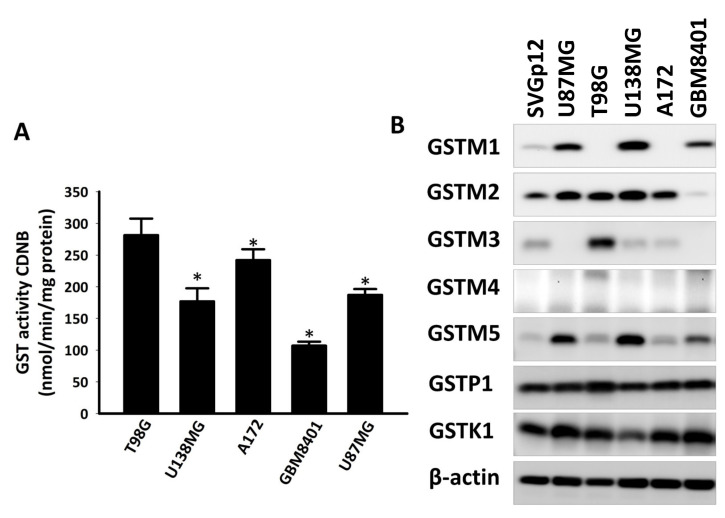
Glutathione S-transferase (GST) activity and GST subfamily member protein expression in glioblastoma cell lines. (**A**) The reaction mixture contained 100 µg of cytosolic protein of each glioblastoma multiforme (GBM) cell line. Values are expressed as mean ± SD (*n* = 3). * *p* < 0.05 relative to T98G cells. (**B**) Western blot analysis of GSTM1, GSTM2, GSTM3, GSTM4, GSTM5, GSTP1, and GSTK1 of human fetal glial cell line SVGp12 and human glioblastoma cell lines U87MG, T98G, U138MG, A172, and GBM8401. β-Actin was used as the loading control.

**Figure 3 ijms-22-07080-f003:**
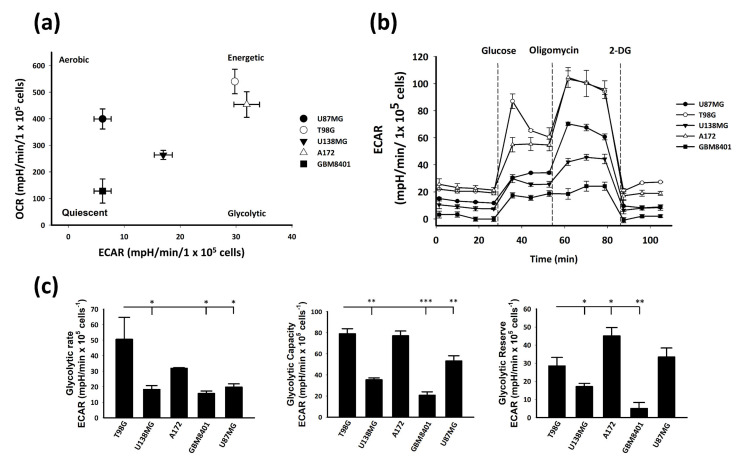
Bioenergetic profiles of five well-known GBM cell lines. (**a**) Basal extracellular acidification rate (ECAR) and oxygen consumption rate (OCR) levels provide a snapshot of the bioenergetics profile of well-known GBM cell lines U87MG, T98G, U138MG, A172, and GBM8401. Values were measured and normalized to cell numbers by using the Seahorse Extracellular Flux (XF) Analyzer. (**b**) Real-time measurement of the ECAR as the glycolysis rate. Glucose (10 mM), oligomycin (1 µM) and 2-DG (50 mM) were injected at the indicated times. Values are expressed as mean ± SD (*n* = 3). (**c**) The glycolytic rate, glycolytic capacity, and glycolytic reserve were analyzed using standard methods. Values are expressed as the mean ± SD (*n* = 3). * *p* < 0.05, ** *p* < 0.01, *** *p* < 0.001 relative to T98G cells.

**Figure 4 ijms-22-07080-f004:**
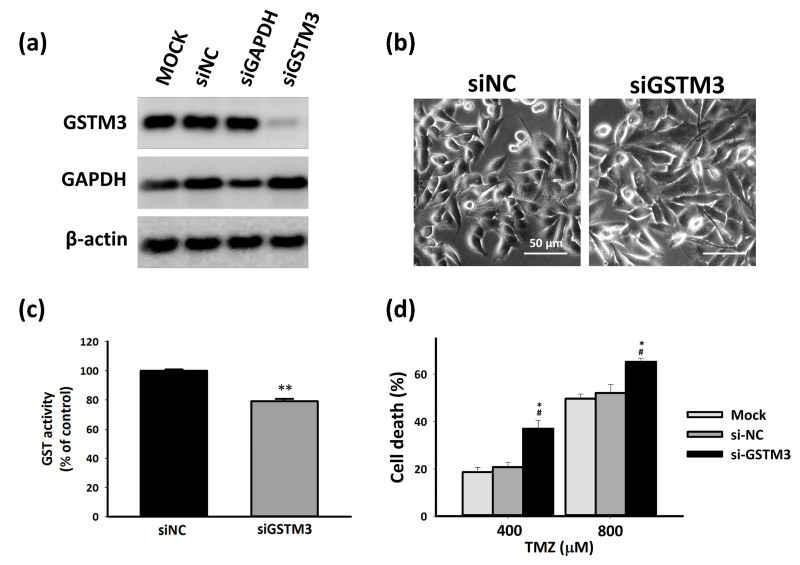
Knockdown of GSTM3 T98G cells. (**a**) Protein expression of GSTM3 and glyceraldehyde-3-phosphate dehydrogenase (GAPDH) in siNC and siGSTM3 T98G cells. β-Actin was used as the loading control. (**b**) Morphology of small interfering RNA (siRNA) negative control (siNC) and GSTM3 (siGSTM3) T98G glioblastoma. Cells were subjected to microscopy at 40× magnification. (**c**) GST activity of siNC and GSTM3 T98G cells. The reaction mixture contained 100 µg of cytosolic protein in 0.1 M KPO4 buffer, pH 7.5, containing 1 mM GSH and 1 mM CDNB at 25 °C. Values are expressed as mean ± SD (*n* = 3). ** *p* < 0.001 relative to siNC. (**d**) MTT assays were performed on Mock, siNC, and GSTM3 T98G cells. The MTT assay of T98G cells after 72 h of TMZ treatment. Values are expressed as mean ± SD (*n* = 6). * *p* < 0.05 relative to siNC. # *p* < 0.05 relative to mock cells.

**Figure 5 ijms-22-07080-f005:**
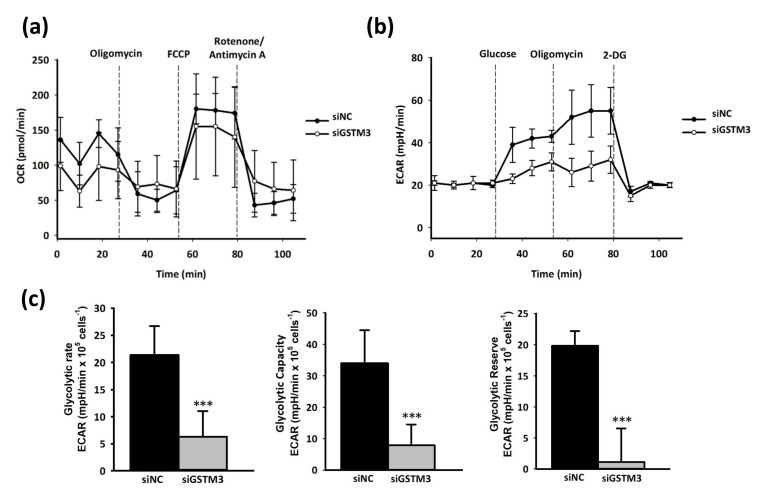
OCR and ECAR values from a glycolysis stress test in GSTM3-knockdown T98G cells. T98G cells treated with siRNA knockdown were depleted 72 h after transfection. The ECAR was measured and normalized to cell number by using the Seahorse XF Analyzer in cells. (**a**) Real-time measurement of the OCR as mitochondrial respiration. Oligomycin (1 µM), FCCP (0.5 µM), and rotenone/antimycin A (0.5 µM) were injected at the indicated times. Values are expressed as mean ± SD (*n* = 3). (**b**) Real-time measurement of the ECAR as the glycolysis rate. Glucose (10 mM), oligomycin (1 µM), and 2-DG (50 mM) were injected at the indicated times. Values are expressed as mean ± SD (*n* = 3). (**c**) The glycolytic rate, glycolytic capacity, and glycolytic reserve were analyzed using standard methods. Values are expressed as the mean ± SD (*n* = 3). *** *p* < 0.001 relative to siNC.

**Figure 6 ijms-22-07080-f006:**
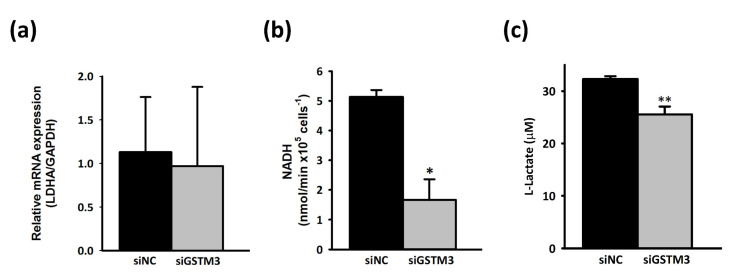
Knockdown of siGSTM3 affects LDH activity in T98G glioblastoma cells. T98G cells treated with siRNA knockdown were depleted 72 h after transfection. (**a**) The lactate dehydrogenase A (LDHA) mRNA expression level was examined using reverse-transcription polymerase chain reaction. GAPDH was used as an internal control. Values are expressed as the mean ± SD (*n* = 3). * *p* < 0.05 relative to siNC. (**b**) The LDH activities of siNC and siGSTM3 T98G cells were measured using a Pierce LDH Cytotoxicity Assay Kit (Thermo Scientific). Endogenous proteins were extracted from each cell population. Each measurement was obtained for 1 × 10^5^ cells. Values are expressed as the mean ± SD (*n* = 3). * *p* < 0.05 relative to siNC. (**c**) The L-lactate content of siNC and siGSTM3 T98G cells was measured using an L-Lactate Assay Kit. Each measurement was obtained for 100 µg of lysate. Values are expressed as the mean ± SD (*n* = 3). ** *p* < 0.01 relative to siNC.

**Figure 7 ijms-22-07080-f007:**
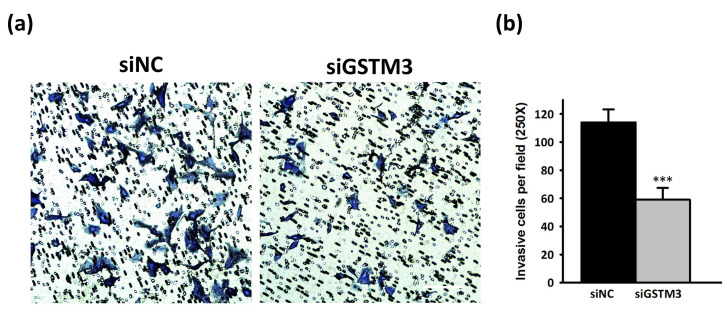
GSTM3 downregulation decreases the invasion ability of T98G glioblastoma cells. (**a**) Cell migration was measured using a collagen-I-coated Transwell chamber (8-µm pore) for evaluating the migration of siNC and GSTM3 siRNA-treated cells. Migrated cells were stained with Giemsa solution (magnification 200×). (**b**) Cells on the underside of the Transwell insert were counted per file. Data are presented as mean ± SD (*n* = 3). *** *p* < 0.001 relative to siNC.

**Figure 8 ijms-22-07080-f008:**
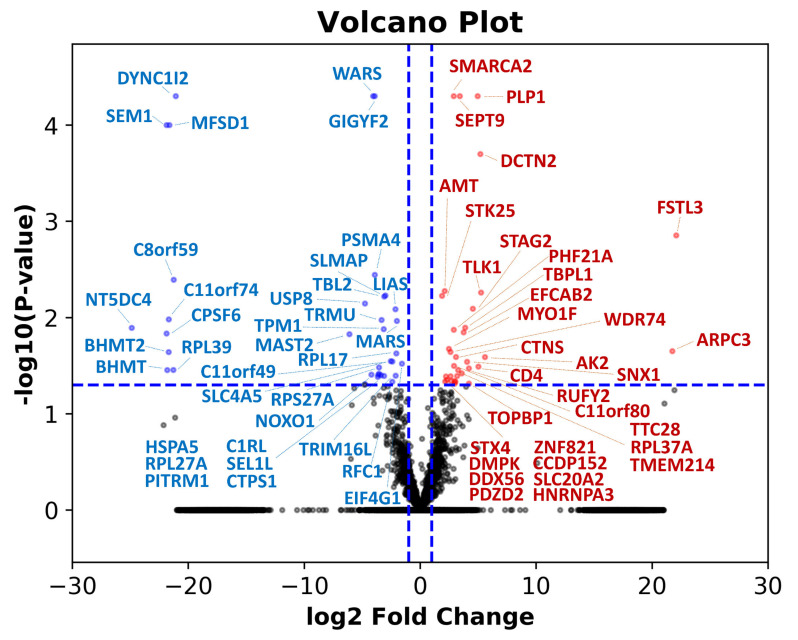
Differential gene expression patterns between GSTM3 inhibited or negative control of T98G glioblastoma cells. The volcano plot of log2 (fold change) versus −log10 (*q*-value) reveals differentially upregulated (right upper quadrant) and downregulated (left upper quadrant) genes expressed in GSTM3-inhibited cells versus negative control cells. Genes with q-values >0.05 and >2-fold changes are plotted in blue (downregulated) and red (upregulated), respectively.

**Table 1 ijms-22-07080-t001:** The upregulated mRNAs revealed in GSTM3 knockdown T98G.

Gene Name	FPKM		Log2 Ratio (siGSTM3/siNC)	*p*-Value
	T98G siNC	T98G siGSTM3		
FSTL3	0.0001	44.3991	18.76017	1.40 × 10^−3^
ARPC3	0.0001	35.1572	18.42346	2.24 × 10^−2^
AK2	2.8863	138.5150	5.58467	2.58 × 10^−2^
TLK1	3.2517	123.6180	5.24854	5.50 × 10^−3^
DCTN2	0.9305	33.9071	5.18751	2.00 × 10^−2^
CD4	1.7614	57.1427	5.01978	3.24 × 10^−2^
PLP1	0.7588	23.5390	4.95525	5.00 × 10^−5^
STAG2	1.4932	34.7032	4.53858	8.10 × 10^−3^
TOPBP1	1.4513	26.6996	4.20137	4.86 × 10^−2^
RUFY2	1.4554	26.7593	4.20055	3.38 × 10^−2^
SNX1	1.4584	24.0448	4.04325	2.89 × 10^−2^
TBPL1	38.1513	562.6220	3.88236	1.28 × 10^−2^
EFCAB2	2.8386	38.0916	3.74624	1.43 × 10^−2^
TTC28	2.0874	25.0200	3.58328	3.81 × 10^−2^
SMARCA2	4.7047	50.2363	3.41655	5.00 × 10^−5^
RPL37A	3.4274	33.1279	3.27288	3.52 × 10^−2^
TMEM214	3.5254	32.0400	3.18400	4.05 × 10^−2^
CTNS	2.7781	23.5403	3.08294	2.57 × 10^−2^
CCDC152	82.5357	677.1110	3.03630	4.66 × 10^−2^
PDZD2	3.8960	30.6379	2.97524	4.86 × 10^−2^
SLC20A2	4.8062	37.6060	2.96799	4.56 × 10^−2^
C11orf80	4.0434	30.4559	2.91308	3.18 × 10^−2^
PHF21A	4.2593	31.6749	2.89465	1.34 × 10^−2^
9-Sep	3.2125	23.8259	2.89076	5.00 × 10^−5^
DMPK	3.7508	23.8481	2.66859	4.56 × 10^−2^
HNRNPA3	71.1478	436.2750	2.61635	4.08 × 10^−2^
WDR74	23.1559	141.5240	2.61159	2.28 × 10^−2^
MYO1F	24.2133	134.5230	2.47398	2.12 × 10^−2^
DDX56	5.8580	28.8715	2.30117	4.45 × 10^−2^
ZNF821	5.7829	26.9271	2.21919	4.10 × 10^−2^
STX4	4.9308	21.8556	2.14811	4.62 × 10^−2^
AMT	8.8386	38.2299	2.11281	5.30 × 10^−3^
STK25	7.1624	26.4060	1.88235	5.95 × 10^−3^

**Table 2 ijms-22-07080-t002:** The downregulated mRNAs revealed in GSTM3 knockdown T98G.

Gene Name	FPKM		Log2 Ratio (siGSTM3/siNC)	*p*-Value
	T98G siNC	T98G siGSTM3		
NT5DC4	308.5000	0.0001	−21.55684	1.28 × 10^−2^
CPSF6	38.4843	0.0001	−18.55391	1.47 × 10^−2^
MFSD1	38.4388	0.0001	−18.55220	1.00 × 10^−4^
BHMT	36.8784	0.0001	−18.49242	3.52 × 10^−2^
BHMT2	33.8849	0.0001	−18.37028	2.29 × 10^−2^
C11orf74	33.6801	0.0001	−18.36154	1.05 × 10^−2^
SEM1	32.3928	0.0001	−18.30531	1.00 × 10^−4^
RPL39	25.6019	0.0001	−17.96589	3.51 × 10^−2^
C8orf59	25.0812	0.0001	−17.93625	4.05 × 10^−3^
DYNC1I2	22.1868	0.0001	−17.75934	5.00 × 10^−5^
MAST2	43.2274	0.6275	−6.10608	1.49 × 10^−2^
USP8	26.5156	0.9751	−4.76518	7.15 × 10^−3^
SEL1L	37.3222	1.3826	−4.75460	4.92 × 10^−2^
NOXO1	94.0989	5.0895	−4.20860	3.92 × 10^−2^
WARS	57.2594	3.4224	−4.06443	5.00 × 10^−5^
GIGYF2	23.4590	1.5490	−3.92074	5.00 × 10^−5^
SLMAP	26.5011	1.7544	−3.91700	3.60 × 10^−3^
CTPS1	22.0086	1.7447	−3.65706	4.14 × 10^−2^
C1RL	39.4514	3.2263	−3.61214	3.94 × 10^−2^
RPL27A	38.0491	3.2439	−3.55207	3.83 × 10^−2^
RPS27A	24.8043	2.1152	−3.55174	3.30 × 10^−2^
HSPA5	225.3150	21.6416	−3.38006	3.98 × 10^−2^
TRMU	66.6088	6.6138	−3.33216	1.06 × 10^−2^
TPM1	24.7276	2.7889	−3.14836	1.32 × 10^−2^
PITRM1	33.4715	3.8571	−3.11733	4.05 × 10^−2^
TBL2	26.9416	3.1387	−3.10161	6.05 × 10^−3^
PSMA4	26.1531	3.2725	−2.99851	5.90 × 10^−3^
SLC4A5	37.3378	6.4214	−2.53968	2.84 × 10^−2^
RFC1	31.6228	5.9547	−2.40887	4.65 × 10^−2^
C11orf49	43.9205	8.3097	−2.40203	2.87 × 10^−2^
LIAS	42.5255	9.6781	−2.13554	8.20 × 10^−3^
TRIM16L	28.4544	6.5612	−2.11661	4.02 × 10^−2^
RPL17	28.1006	6.7387	−2.06005	2.36 × 10^−2^
MARS	92.0487	22.8211	−2.01203	1.09 × 10^−2^
EIF4G1	25.3945	8.5137	−1.57666	3.02 × 10^−2^

## Data Availability

The datasets generated during and/or analyzed during the current study are available from the corresponding author on reasonable request.

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
