# Peer review of "Glutathione S-Transferase M3 Is Associated with Glycolysis in Intrinsic Temozolomide-Resistant Glioblastoma Multiforme Cells"

_ijms, 2021, doi:10.3390/ijms22137080_

Round 1

Reviewer 1 Report

I congratulate authors for presenting a well written and scientifically sound manuscript describing the role of Glutathione S-transferease (GST) isoenzymes especially GSTM3 in TMZ resistant GBM cells.

GSTM3' role in GBM (to the best of my knowledge) is still unexplored and therefore, this manuscript presents a novel report on this subject. 

The GST subfamily of isoenzymes including GSTM3 are supposed to detoxify the cells from harmful agents, however a growing body of evidences  demonstrating their role in tumour development and drug-resistance is particularly interesting.   

TMZ resistant cell culture (T98G) used in the current study had a specific and high protein expression of GSTM3 which when knocked down using siRNA clearly demonstrated its role in TMZ toxicity, glycolysis and cell migration.

As a reviewer, i would recommended and prefer to see more than one GBM cell culture in the study to demonstrate GSTM3' role in TMZ drug induced resistance. I also suggest the authors to elaborate the current studies to incorporate more GBM biopsy derived cell cultures, to increase the significance of the current study.

I particular found use of seahorse extracellular flux analysis unique and interesting in the current report. The bioenergetic profile of the cells adds an additional measure of confirmation to the already known metabolically skewed tumour cells T98G and A172 [Fig3 (a), (b) and (c)]. 

Best wishes,

Author Response

We would like to thank you for carefully reviewing our work and for your invaluable suggestions. We evaluated in 5 GBM cell lines the TMZ treatment responses and intrinsic GSTM3 expressions. T98G and U138MG are TMZ resistant GBM cell lines widely used in drug resistant brain cancer research, while the other three are more sensitive to TMZ treatment. As shown in Figure 2(b), our results indicated that GSTM3 was highly expressed in T98G, but not so much in the U138MG cell line. Moreover, according to the data, each GBM cell line showed different expression levels among different GSTM subfamilies. Therefore, T98G was chosen as our top candidate to explore the role of TMZ resistant GBM cells in tumor development for this report. We are continuing to further explore the role of different subfamilies of GSTM proteins in chemo-resistant brain cancer. Thank you again for your insightful opinions.

Reviewer 2 Report

The authors of the manuscript “GSTM3 Is Associated with Glycolysis in Intrinsic Temozolomide-Resistant Glioblastoma Multiforme Cells” provide strong evidence that GSTM3 is involved not only in detoxification but also regulates the activity of lactate dehydrogenase A and modulate the glycolysis in cancer that provides the chemoresistance to the cells. The data are well documented and confirmed by different approaches. The findings provided in the manuscript illustrate the idea that cancer cells use multiple mechanisms to improve their viability and will be interesting to the wide auditorium.

However, the manuscript needs to be improved before acceptance. It is not clear how it was calculated the glycolytic rate, glycolytic capacity, and glycolytic reserve demonstrated in Figures 3c and 5c. The description of the methods used for data obtaining and calculations must be included in the manuscript.

Author Response

We would like to thank you for taking the time to review our work and for your invaluable suggestions. Based on your comments, we have made some changes to the manuscript, which are detailed below.

We have added the supplier information to Methods, with the sentence as follows:Equations of glycolytic rate, glycolytic capacity, and glycolytic reserve were referenced from Agilent Seahorse XF Glycolysis Stress Test Report Generator User Guide. Those were calculated using the following formula:

Glycolytic rate = (Maximum rate measurement before Oligomycin injection) – (Last rate measurement before Glucose injection)

Glycolytic Capacity = (Maximum rate measurement after Oligomycin injection) – (Last rate measurement before Glucose injection)

Glycolytic reserve = (Glycolytic Capacity) – (Glycolytic rate).
